# A Modular In-Vehicle C-ITS Architecture for Sensor Data Collection, Vehicular Communications and Cloud Connectivity

**DOI:** 10.3390/s23031724

**Published:** 2023-02-03

**Authors:** David Rocha, Gil Teixeira, Emanuel Vieira, João Almeida, Joaquim Ferreira

**Affiliations:** 1Instituto de Telecomunicações, Universidade de Aveiro, Campus Universitário de Santiago, 3810-193 Aveiro, Portugal; 2Departamento de Eletrónica, Telecomunicações e Informática, Instituto de Telecomunicações, Universidade de Aveiro, Campus Universitário de Santiago, 3810-193 Aveiro, Portugal; 3Instituto de Telecomunicações, Escola Superior de Tecnologia e Gestão de Águeda, Universidade de Aveiro, Campus Universitário de Santiago, 3810-193 Aveiro, Portugal

**Keywords:** cooperative intelligent transport systems, in-vehicle sensor data collection, vehicular communications systems, mobile network technologies

## Abstract

The growth of the automobile industry in recent decades and the overuse of personal vehicles have amplified problems directly related to road safety, such as the increase in traffic congestion and number of accidents, as well as the degradation of the quality of roads. At the same time, and with the contribution of climate change effects, dangerous weather events have become more common on road infrastructure. In this context, Cooperative Intelligent Transport Systems (C-ITS) and Internet of Things (IoT) solutions emerge to overcome the limitations of human and local sensory systems, through the collection and distribution of relevant data to Connected and Automated Vehicles (CAVs). In this paper, an intra- and inter-vehicle sensory data collection system is presented, starting with the acquisition of relevant data present on the Controller Area Network (CAN) bus, collected through the vehicle’s On-Board-Diagnostics II (OBD-II) port, as well as on an on-board smartphone device and possibly other additional sensors. Short-range communication technologies, such as Bluetooth Low Energy (BLE), Wi-Fi, and ITS-G5, are employed in conjunction with long-range cellular networks for data dissemination and remote cloud monitoring. The results of the experimental tests allow the analysis of the road environment, as well as the notification in near real-time of adverse road conditions to drivers. The developed data collection system reveals itself as a potentially valuable tool for improving road safety and to iterate on the current Road Weather Models (RWMs).

## 1. Introduction

Road traffic has experienced constant growth in recent decades around the world. As a result of the increase in the number of vehicles in circulation, concerns have arisen about road safety. Despite the efforts of manufacturers to offer more advanced and effective safety triggers in their vehicles, more than 1 million fatalities happen every year globally, according to the World Health Organization [1]. Traffic congestion is also a problem [2] and drivers spend a significant amount of time in traffic queues every day in big cities all over the world. This has consequences on public health, quality of life, and economic costs. Air pollution is also now a major concern for automotive manufacturers.

Combining all negative aspects of the road environment, a need for the development of new technologies not only in vehicles but also in infrastructure arises in order to mitigate driver’s limitations such as slow reaction times and reduced line of sight, providing them and their cars the tools to become more aware of their surroundings and more capable of preventing dangerous situations. By using both short-range and long-range communications technologies among vehicles and between vehicles and the infrastructure, a connected environment can be achieved, creating a large network of vehicular nodes and providing life to the term C-ITS. Solutions already exist where not only vehicles with communication capabilities are integrated, but also legacy vehicles that cannot communicate with other parties, creating a solid base for these technologies emerge in real traffic scenarios. Autonomous vehicles also extend the conventional vehicle’s sensor data to include other sources [3]. This can be used to improve road safety by integrating the sensors with a data collection mechanism that uses Vehicle-to-Everything (V2X) to broadcast dangerous road conditions to neighboring road users.

When sensor data from vehicle-integrated sensor systems and external sensors become available, an improved perception of the surrounding environment can be provided. In addition to that, access to in-vehicle data is increasingly desired by stakeholders in the automotive ecosystem and related sectors, with great potential to be monetized [4,5]. This work directly contributes to this extended perception of the traffic environment by proposing a modular and scalable C-ITS architecture to be deployed in vehicular systems. It integrates the collection of in-vehicular sensor data from the CAN bus, the dashboard smartphone, and other added car sensors with localized vehicular communications, long-range cellular networks, and cloud connectivity. As a result, data dissemination regarding the road environment and more detailed vehicle statuses can be implemented, enabling real-time warning notifications to vehicles and cloud monitoring of the road traffic system.

The rest of the paper is organized as follows. Section 2 focuses on the background and related work of vehicles’ sensor collection and data dissemination through vehicular networks. Then, Section 3 describes the equipment and methods used for identifying vehicular sensors’ parameters in the CAN bus. After that, Section 4 presents the proposed architecture of the in-vehicle C-ITS system, which includes the integration of the vehicle’s sensors data with the ITS-G5 protocol stack and the cloud communications framework. Section 5 describes the implementation details in the On-Board Unit (OBU), as well as on the other components such as the OBD-II reader and the smartphone device. The performed tests and evaluation results used to validate the proposed system are presented in Section 6. Finally, Section 7 concludes the paper with a summary of its contributions and main conclusions.

## 2. Related Work

Two important topics were analyzed for the state-of-the-art review regarding the vehicular dissemination of in-vehicle sensor data. Firstly, the relevant sensor data that can be extracted from in-vehicle systems were identified, as well as the data collection methods and their respective integration in RWMs. Then, innovative road awareness and perception mechanisms were studied in the context of C-ITS architectures based on V2X communications.

### 2.1. Vehicular Sensor Data

The success of C-ITS largely depends on the platform used to access, collect, and process accurate data from the environment. Guerrero-Ibaze et al. [6] state that integrating sensors with the current transportation systems allows the mitigation of some of the problems that past and current transportation systems have been facing, such as high levels of CO${}_{2}$ emissions and high levels of traffic congestion. Additionally, the author discusses how user applications can benefit from sensor technologies embedded in Intelligent Transport Systems (ITS) components, which enhances the experience of various functionalities, namely driver assistance, collision alerts, and traffic control.

Inferring events or statistics from the road environment from vehicle-integrated sensor systems has a been topic of study by various authors. The comparison of each wheel speed, the position of the pedals, and the speed of the vehicle can be used to detect slippery road conditions in real-time [7,8], as well as pavement damage [9]. Adding external sensors such as the ones present in smartphones can also be combined with the vehicle’s own sensors to add redundancy and improve the accuracy of these detection mechanisms. Yunfei et al. [10] use the smartphone’s inertial sensors as well as OBD-II data, without tapping into the proprietary data, to detect slippery road conditions. Other uses of these data include the evaluation of road weather conditions [11,12,13], using the status of sensors such as windshield wipers and headlights, and remote vehicle diagnostic reporting [14] to relay malfunctions. As an effort to improve safety, vehicular data allow for the assessment of adequate driving behaviors [15] in specific road–weather conditions, causing it to be possible to send recommendations to the user [16] based either on dangerous driving or irresponsible fuel consumption.

Recent years have seen an increased focus on tire–road friction research, as it constitutes a vital aspect for determining the effectiveness of active safety systems and minimizing the likelihood and impact of traffic accidents [17,18]. Pomoni [19] explores the benefits and limitations of smart tires that are able to measure the friction of the road surface, enabling the tracking of pavement skid resistance in real time. The author mentions connectivity as a key factor when developing systems that are embedded in tires, as it is expected to be a dominant aspect of future mobility, enabling other stakeholders such as road managers or other vehicles to potentially access the data. The same work concludes that incorporating adhesive sensors within the tires is a cost-efficient method for implementing smart tires and that this approach is likely to aid drivers in adjusting their behavior in response to indications of reduced skid resistance, thereby enhancing road safety.

Bogaerts et al. [20] propose a model to enhance the current RWM by using vehicular sensor data and to enable real-time road weather warnings for local weather phenomena and dangerous road conditions. Vehicles are equipped with a CAN reader and external sensors and the data collected by the fleet are then sent to a cloud back-end using a data distribution framework. Although CAN data were collected, within this study, only data from external sensors were used to investigate the relationship between measurements and road conditions, since access to CAN parameters is not always available and can be erroneous due to the use of reverse engineering processes. Table 1 overviews the external sensors used in this study.

Every 3 s, when the Global Positioning System (GPS) module presented a new position value, the sensor measurements were collected in the current timestamp, as well as all the CAN messages within this time interval, and then sent to a data distribution platform. With the data from around 2000 h of driving collected from 15 vehicles, the authors’ findings indicated the potential value of vehicle sensors in relation to road weather. With these sensor measurements, the patterns in the temperature values were correlated with the instants when the rain started or stopped. The results showed a median Root-Mean-Square Error (RMSE) close to 4 °C between the temperature forecasts performed at a Road Weather Station (RWS) and the ones performed at close observed road segments.

Following that and using the same equipment as in [20], Bogaerts et al. [21] demonstrated two examples of how artificial intelligence and machine learning can be leveraged to process data from a fleet of vehicles and obtain relevant weather information with an improved data quality compared to the raw measurements, using convolution neural networks to extract visibility and precipitation data from raw camera images. With this, a new RWM was presented to enrich the modeling of road weather with car data at a high spatiotemporal resolution. The results showed a median RMSE below 2 °C between the temperature forecasts performed at RWS and those performed at the closely observed road segments close.

### 2.2. Road Perception

While data collection from the road traffic environment has been technologically evolving, the current trend in Road Weather Information Systems (RWIS) is also in the direction of greater integration with all road agents, namely connected vehicles and Vulnerable Road Users (VRUs). Low latency V2X communications between vehicles, the roadside infrastructure, and cloud systems can be used for the quick local dissemination of weather warnings [22]. For instance, a variable speed limit system can be implemented depending on weather conditions, in which the maximum speed values are disseminated through vehicular networks [23]. The 5G-SAFE-PLUS project is currently integrating 5G communications in the road infrastructure, in order to enable the fast exchange of data with vehicles and the transmission of real-time weather warnings [24].

Similarly, ensuring road users are mindful of their surroundings is essential in enabling safer conditions for them, while much significant work is performed with drivers in mind, VRU entities such as pedestrians also represent a significant portion of the road environment and may be the most neglected, with very few ways of protecting themselves during accidents.

Willecke et al. [25] mention the importance of raising the awareness of VRU objects to both drivers and these users, which can be achieved through the Collective Perception Service (CPS) defined by European Telecommunications Standards Institute (ETSI). To evaluate the impact of the CPS service on VRUs, the authors employ a simulation using *Monaco SUMO Traffic (MoST)* as a scenario. Here, V2X-enabled vehicles are equipped with two radar sensors to mimic the sensing capabilities of modern vehicles. The first sensor is attached to the front bumper while the second is attached to the rear bumper. Therefore, equipped vehicles have omnidirectional perception with a range, similar to the perception capabilities used for the Tesla Autopilot. The evaluation investigated the influence of VRUs on various metrics. Although the channel load induced by their inclusion was little, the number of perceived objects increased significantly, where almost half of the objects included in Collective Perception Message (CPM) were persons. The safety benefit was measured by counting the amount of VRU objects perceived in the direct vicinity of vehicles. The inclusion of CPS improved vehicle awareness and detection delays. The study concludes that CPS will play an important role in the adoption of V2X and, while much work is being performed on this topic, most of it is focused on vehicles, despite the inclusion of VRUs in the key use-cases of collective perception.

In addition to the CPS, which enables the passive perception of VRUs, VRU Awareness Service (VAS) can also be actively used to leverage their perception by means of V2X communication with the transmission of VRU Awareness Message (VAM)s. Lobo et al. [26] compare the usage of these services both in separate and combined form through the representative roundabout in the *Ingolstadt traffic scenario (InTAS)*. When combining both services, the best perception rate was achieved, in which an average of 6% more VRUs were detected in comparison to the second best (only CPS). This also resulted in the shortest detection time. With this method, the maximum channel busy ratio was increased by 18%. The study concludes that, to improve VRUs safety, both services should be used, providing a better perception environment to each user, while not overloading the communication channels.

The work described in detail in the following sections uses the knowledge shared by these authors to build a reliable mechanism capable of extracting vehicular sensor data and disseminating it to other ITS entities, while also using a mobile application that leverages its user’s road perception.

## 3. Materials and Methods

In this work, a set of specific hardware and reverse engineering methods were used to extract data from the vehicle CAN bus and correctly identify the intended sensor parameters. This section presents the equipment installed inside the probe vehicle for sensor data collection together with algorithms applied to the CAN messages wiretapped from the OBD-II port.

Concerning the hardware equipment, a *Carloop* with a *RedBear Duo* combo is used as a CAN transceiver and controller, respectively. The microcontroller includes both WiFi and BLE interfaces, serving as an OBD-II reader that wirelessly connects to a vehicle’s OBU. This OBU is based on a board from *PC Engines* with a set of extensions, being the main hardware specifications detailed in Table 2. Figure 1 and Figure 2 show the OBD-II reader kit and the OBU used.

With respect to the reverse engineering methods, the algorithms used in this work for CAN bus parameter identification are divided into binary and non-binary methods.

### 3.1. Binary Parameters

For binary parameters, such as the headlight status, windshield wiper state, or pedal status, the identification can be performed through manual interaction with each car function [27]. The following phases are defined:Reference—In this phase, the user needs to not perform any action inside the vehicle for a while. The algorithm searches for bits that did not show any transition, saving the reference for the next phase;Monitoring—Here, the user will have the order to perform a specific number of transitions in the parameter he wants to identify (e.g., switching on and off the minimum headlights). The algorithm then filters the bits with a bit-flip count of twice the number of triggers and deletes those that present different results. These filtered parameters are the only ones that the algorithm will be aware of in the next phase;Validation—This phase serves as confirmation to ensure that the identification is correct.

### 3.2. Non Binary Parameters

These are based on an offline algorithm that uses previously acquiredCAN traces and calculates the Pearson’s correlation coefficient between different CAN messages and other sources, such as the OBD-II. Given a pair of variables (x,y), the correlation ρ is provided by:(1)$$\rho =\frac{\sum_{i=1}^{n}\left(x_{i}-\bar{x}\right)\left(y_{i}-\bar{y}\right)}{\sqrt{\sum_{i=1}^{n}{\left(x_{i}-\bar{x}\right)}^{2}{\left(y_{i}-\bar{y}\right)}^{2}}}$$

Different methods exist for correlating data between different data sources, but the one described takes into account factors such as different offsets and scales.

However, this algorithm assumes that *x* and *y* are arrays of the same size, but because different CAN messages do not have the same sampling frequency, the data must be interpolated to achieve the same number of samples in both variables. Given two points $\left(x_{1},y_{1}\right)$ and $\left(x_{2},y_{2}\right)$, an interpolated value (x,y) can be obtained using the linear interpolation method:(2)$$y=y_{1}+\left(x-x_{1}\right)\frac{\left(y_{2}-y_{1}\right)}{\left(x_{2}-x_{1}\right)}$$

For demonstration purposes, a trip can be carried out while always creating different OBD-II requests through the CAN controller, sending every captured CAN message to an application that is listening and logging all the messages. After acquiring the trace, the following steps are carried out to identify the CAN parameters (Figure 3 illustrates this process):Pre-Processing—In this step, every byte from different CAN IDs is grouped in a data structure. This results in various arrays containing the evolution of different bytes throughout the trace. Every pair $\left(ID,Byte_{index}\right)$ becomes assigned to one of these arrays;Processing—In this step, the algorithm will extract a reference array from an input and calculate its correlation to every other $\left(ID,Byte_{index}\right)$ pair;Validation—Finally, the algorithm outputs the pair $\left(ID,Byte_{index}\right)$ that most resembles the input assigned, with its correlation coefficient.

## 4. Architecture

Figure 4 shows the devices installed inside the vehicle. A CAN/OBD-II reader is used to extract relevant data from the CAN bus. The smartphone is placed on the dashboard, which collects data from sensors, such as luminosity, barometric pressure, and others. Lastly, the ITS-G5/Long Term Evolution (LTE) station has the responsibility to communicate with other vehicles, the roadside infrastructure, and the cloud (either directly using short-range technologies or through cellular networks). On the intra-vehicle network, every device exchanges data via wireless technologies (BLE or WiFi). The ITS-G5 station also has the capacity to obtain data from an external Global Navigation Satellite System (GNSS) module, located in the OBU.

The collected data are sent to a cloud platform, in which a Message Queuing Telemetry Transport (MQTT) broker is used for real-time message distribution, providing an endpoint for vehicles to publish and subscribe to different topics in order to share live data to other vehicles in motion. Other cloud resources include logging services, databases for storage, and a Web Application Programming Interface (API) that offers users access to historic data.

### 4.1. Vehicle Sensor Data

Figure 5 summarizes the data collection process. Each vehicle contains many control units that communicate through messages on the CAN bus. These messages are read through a micro-controller attached to the vehicle OBD-II port, which periodically disseminates the relevant data to the ITS-G5 OBU using BLE. In detail, three components are defined here:**Vehicle (1)**: Integrates an internal network based on the CAN protocol, which serves as a medium for monitoring and controlling various parts of vehicles through each Electronic Control Unit (ECU). An OBD-II standardized port is present in every vehicle that allows for tracing these messages;**CAN/OBD-II Reader (2)**: A previous identification of relevant messages is needed to know what to listen to. Many of these messages obey proprietary protocols, which causes it to be hard to acquire every intended parameter. The chosen device for this was a Carloop with a RedBear Duo microcontroller, which contains Universal Serial Bus (USB), BLE, and WiFi communication interfaces;**ITS-G5/LTE Platform (3)**: Responsible for vehicular communications with roadside infrastructure, other vehicles in its vicinity and the cloud. This unit is used as an OBU, based on a platform from *PC Engines* that includes a WiFi module with an Atheros driver adapted for ITS-G5, a LTE module, and an external GPS receiver. Communication between this platform and the CAN reader is performed via the BLE interface.

### 4.2. Smartphone Sensor Data

In addition to the collection of data from the CAN bus, a smartphone is also included in this architecture, being installed in the dashboard of the vehicle in order to capture additional environmental data. The communication between the smartphone and the OBU is performed via WiFi. To accomplish this, an access point is created at the OBU by using an USB WiFi dongle that manages the local wireless network. Furthermore, a mobile application is also present with the goal of providing a user interface to the driver and passengers of the vehicle. The app improves the perception of the current road environment by displaying road and meteorological warnings to the user. Figure 6 represents a block diagram of the application architecture, showing the external components to which it is connected and the internal sensors from which it collects data at a configurable rate.

In order to allow, with a singular code base, the generation of binaries for both Android and iOS, the application was developed using *React Native*. For interaction with the smartphone’s sensors, *Expo* is used, which is an API/framework written in JavaScript, that includes implementations of many APIs common to both platforms and exposes them to *React Native*, allowing the code base to be written only in Typescript/Javascript. *Expo-Sensors* were used for sensor data collection to which the authors directly contributed by providing a Light Sensor implementation for Android ([28]). Finally, React native maps were chosen for the visualization and reporting of events, presented as map markers and/or in-app notifications, through various messages defined by ETSI, such as Decentralized Environmental Notification Messages (DENMs), Cooperative Awareness Messages (CAMs), Infrastructure to Vehicle Information Messages (IVIMs), VAMs, CPMs, and other proprietary messages such as Heartbeats, which are used to monitor the deployed road infrastructure.

### 4.3. Platform Internal Architecture

Figure 7 represents the organization of the various services installed in the OBU, how they interact with the wireless communication endpoints, and the data exchanges that exist. Three new types of messages are defined:**OVSM**: OBD-II’s Vehicular Sensor Message—formed with an ETSI ITS standard header and a message body, including the reference position and the sensor data that are available from the OBD-II reader.**SPVSM**: SmartPhone’s Vehicular Sensor Message—includes an ETSI ITS standard header and a message body with both the reference position and the sensor data available from the smartphone device;**VSM**: Vehicular Sensor Message—combines the content of the previous messages (whenever available) with an ETSI ITS standard header.

An MQTT broker acts as the central element that behaves as a message distributor, diffusing messages between the smartphone, the OBU, and other clients. OVSMs and SPVSMs are published here. Different services are then defined to accomplish a scalable and flexible data collection mechanism:**it2s-obd**: Manages the BLE communications between the OBU and the CAN/OBD-II reader, then uses these data to compose the OVSM messages and send them to the MQTT broker.**it2s-peripherals**: Acts as an MQTT client, filters the received messages, such as the SPVSMs and OVSMs, and ensures their data are available to other services via shared memory files. A shared library is included to interact with them.**it2s-gps**: Manages the external GPS module, collects its data, and ensures it is available to the services via a shared memory file. A shared library is included to interact with it.**it2s-data-collection**: Responsible for collecting data from every data endpoint, creating VSM messages, and finally disseminating them to the cloud via cellular networks.

Finally, *ETSI ITS-S Stack* is the implementation of the ETSI-defined ITS Station (ITS-S) reference architecture [29]. The facilities layer is adapted to use the *it2s-peripherals* shared library, using the sensor data to include it in various C-ITS messages such as CAMs or DENMs.

## 5. Implementation

A list of the parameters to collect from each data source (OBD-II, CAN bus, and smartphone) is shown in Table 3.

On the one hand, ideally, the OBD-II parameters would also be collected directly through the CAN bus, because this eliminates the need to send request messages and wait for the responses and also improves the granularity of the collected data. On the other hand, identifying parameters in the non-standardized messages present in the bus is a difficult task, because different vehicle manufacturers develop distinct ways of representing content in CAN messages (e.g., data may be in signed or unsigned values, little-endian or big-endian, and conversion to integers may have different scales and offsets). Because of this, interpreting the data in a generic way is a difficult task. Furthermore, the CAN Database (CAN DBC) files are not easily accessible and the ones that can be found on online forums such as *opengarages* are often incomplete and not available for all car models, causing it to be impossible to have a ground truth set of CAN IDs.

With this in mind, a hybrid solution is implemented. Individual parameter identification was carried out through state-of-the-art reserve engineering methods in order to identify which CAN IDs were relevant, but the results of their application widely vary between different vehicles. This problem is somewhat diminished by the usage of the standardized OBD-II request/response messages, but this is also limited by the parameter IDs defined by the standard.

### 5.1. OBD-II Reader

This entity is implemented through the Carloop development kit that interacts with the CAN bus via the OBD-II port in conjunction with a RedBear Duo development kit that acts as a CAN controller and contains the BLE interface to connect to the ITS-G5 platform.

The data exchanges are defined by the Generic Atribute Protocol (GATT) protocol, which allows communication between devices through the transmission of attributes, consisting of concepts such as services or characteristics. A GATT server is implemented with a custom profile on the BLE peripheral (Carloop), while the BLE central (ITS-G5 platform) initiates the connection and uses the profile to receive data. This profile is configured with a rate of 10 Hz for the device to send a burst of notifications that contain every new reading that was captured from the CAN bus. The payloads of these notifications are composed of the notification ID (NID), which is used to identify which parameter it contains, and four bytes of data, encoded as floating point numbers.

### 5.2. Smartphone Application

The main screen of the mobile application shows a map with the current location of the user, in which other road agents (represented by CAMs, VAMs, or CPMs) and traffic events or information (disseminated via DENM, High-Definition Map (HD-Map), or IVIM messages) are displayed. Three operating modes are implemented: VRU, Traffic Control Center (TCC), and OBU modes.

The VRU mode is intended for pedestrians/cyclists; it connects to the MQTT broker in the cloud through cellular networks and subscribes to the nearby tiles, defined from the quadtree in which the user is currently located. The zoom level used in this tile system is configurable in the settings menu. In this mode, the application sends, at 1 Hz, VAM messages to the central broker in order to disseminate the presence of a vulnerable road user to other ITS stations. The TCC mode is meant for roadside infrastructure operators. In this scenario, the smartphone assumes a *stationId* value of 1 by default, since this is the identifier of the C-ITS central station. In this mode, every message is received. Finally, the OBU mode is for vehicle drivers to acquire an improved perception of the road environment, connecting to the OBU’s MQTT broker and subscribing to all topics. In this mode, the application connects to the OBU’s broker and sends frequent SmartPhone’s Vehicular Sensor Message (SPVSM) messages to it, in order to be later transmitted by the OBU to the cloud.

The application user interface can be visualized in Figure 8, Figure 9, Figure 10, Figure 11, Figure 12 and Figure 13. Figure 8 shows the representation of CPM messages sent from a nearby Roadside Unit (RSU) equipped with a traffic radar (vehicles in blue), as well as CAM messages sent from other vehicles (in green). In Figure 9, a representation of an IVIM is shown, signaling a speed limit of 80 km/h, which is displayed when the user enters the influence area of that traffic information sign. The current distance to the relevant info (in this case, the traffic sign) is also presented. Figure 10 displays the representation of an HD-Map message, including the temporary traffic signs and the identified obstacle on the road. The application also allows for the generation of DENM messages considering many scenarios. Figure 11 shows the menu for the selection of the detected warning event. Finally, Figure 12 presents a tab with the current sensor data of the vehicle, parsed from OBD-II’s Vehicular Sensor Messages (OVSMs). Figure 13 shows another tab with the data extracted from the smartphone sensors, also sent to the local OBU via SPVSMs. This mobile application is available on the Google Play Store (available at https://play.google.com/store/apps/details?id=com.it2s.it2smobileapp, accessed on 1 February 2023) for testing and general validation of its functionality.

### 5.3. Smartphone Sensor Data

While the mobile application’s primary objective is to implement an alert interface for drivers, since vehicles also roam through vast geographical areas, it is beneficial to collect data from its sensors, allowing one to infer the current status of the road environment. As the vehicle automatically collects sensor data, the same can be applied to the mobile application.

One of the biggest causes of road accidents is the lack of visibility, which happens often due to the lack of light on the roads. So, since each smartphone has a luminosity sensor, this can be used to automatically monitor the quantity of light available on the road at every location that the user passes by. This and other smartphone’s sensor data are published to the OBU’s MQTT broker, in the content of a SPVSM, and then to a cloud server in Vehicular Sensor Messages (VSMs). The application also benefits from the smartphone’s gyroscope and accelerometer, regularly collecting data regarding the phone’s rotation and acceleration in each geographical location. Sudden perturbations of these values in the same place by different OBU’s may infer the presence of road deformations or holes. Drivers should pay maximum attention to the driving task, so it is essential that these processes execute in a fully automated way, sending the data to the OBU’s broker, which are then redirected to the cloud.

### 5.4. ITS-G5/LTE Platform

Figure 14 shows the various data flows that exist in the OBU during execution, which are defined by the services that create and process OVSMs, SPVSMs, and VSMs. In detail:


**OVSMs**: Periodically, the Carloop device sends OBD-II requests while listening to every message that is transmitted in the CAN bus and saving the most relevant ones in local registers. A list of relevant CAN IDs and OBD-II PIDs is configured in order to indicate which messages should be sent and listened to. Additionally, at 10 Hz, a burst of notifications is sent to the OBU via BLE, which includes the already decoded values of the sensors where a change in state is observed, followed by a notification signaling the end of the burst, with the payload bits all equal to 1. On the OBU side, the service *it2s-obd* manages the communications with the OBD-II reader, writing the content of the sensor data in local memory and then sending OVSM messages at 1 Hz to the local broker. These messages will then be parsed by the service *it2s-peripherals* and written to a shared memory file;**SPVSMs**: These messages are sent from the smartphone via WiFi to the OBU’s MQTT broker at 1 Hz. They are then received by the service *it2s-peripherals*, which then parses the message, analyses its content, and writes it to a shared memory file;**VSMs**: At 1 Hz, the data from every shared memory file (GPS, OBD-II, and smartphone) is read and included in different VSM body containers, by the *it2s-data-reader* service. Then, the message is sent through the cellular interface LTE/5G to the cloud MQTT broker for remote monitoring of the road environment.


The generation frequencies previously mentioned are the default ones, but each value is independently configurable. On each endpoint the transmissions are asynchronous and its data flow is properly isolated through the shared memory files. These messages are defined by ASN.1 schemas, allowing data structures to be serialized and deserialized in a cross-platform way. The sensor data are located in different containers, which are included in the message bodies of OVSMs and SPVSMs. VSMs then optionally includes each of those bodies, allowing for execution independence between the services; for example, a vehicle may only be equipped with a smartphone and not the OBD-II reader. In this case, the VSMs message only includes the message body of the SPVSMs in addition to the header. The ASN.1 message formats of OVSMs, SPVSMs, and VSMs are presented in Appendix A.

## 6. Tests and Results

In order to validate the full end-to-end vehicular sensor dataflow, several tests were carried out on the streets of Aveiro (Portugal), where ITS-S stations were deployed on the roadside by Instituto de Telecomunicações (IT) [30], as well as during interurban trips in the Aveiro region. These tests were carried out in uncontrolled real traffic scenarios, with varying road geometry features, traffic conditions, and speed limits.

### 6.1. Vehicle Sensor Data

The reverse engineering methods used are mostly based on the monitoring of bitflips for binary parameters and on the correlation with other inputs for non-binary parameters, both online and offline, depending on the need to accommodate the memory limitations of the *Carloop* device.

As an example, the engine speed parameter corresponds to the PID 0x0C in the OBD-II standard, with its value provided by (256A+B)4, where A is the first data byte and B the second. These OBD-II requested values were used to find the best candidates within all of the CAN messages that are read in the bus. The correlation algorithm identified the second and third bytes of the 0xC9 CAN ID as the best candidates, with a very similar evolution compared to the OBD-II data. Figure 15 shows a comparison between both of them.

The relative position of the throttle pedal is also specified as an OBD-II standard parameter. Its value is provided by (100/255)A, where A is the first data byte. The algorithm identified the fifth byte of the 0xC9 CAN ID as the best candidate, even though the scale of the data is inverted. Figure 16 shows a comparison between both of them.

The speed of each wheel can also be easily identified, through the correlation with the vehicle speed when driving in a straight line. To distinguish between the four of them, performing sharp turns while driving results in bigger position circumference lines on the outer wheels than on the inner wheels, which impacts their speed. Figure 17 shows a representation of this. The data about wheel speed values are valuable because they can be used to report aquaplaning events in real-time to other drivers in the vicinity, causing them to be aware of dangerous road conditions as soon as they are detected.

As is clear in the above pictures, the sampling rate of the CAN messages is much higher than what is possible through only OBD-II, since there is no need to wait for the standard request–reply process. However, the identification of every desired parameter is also very difficult to obtain for each distinct vehicle, given the fact that manufacturers might use complimentary buses such as the Local Interconnect Network (LIN) for some sensors or more complex encoding for sensitive CAN messages. Furthermore, a 100% guarantee that the parameters identified are correct is not possible to attain, due to the intrinsic nature of reverse engineering methods. This ultimately removes the option to rely solely on the CAN bus, but its usage as a complement to an already functioning sensor data collection mechanism is valuable.

These results were acquired using an Opel Adam from 2016 and the different offset and scale values were derived to match the corresponding OBD-II data. An outline of the identified parameters and their features is presented in Table 4.

### 6.2. Smartphone Sensor Data

Illuminance is a parameter that varies a lot depending on the scenario. During the day, the sun can radiate directly to the sensor, which leads to a very high illuminance value, or be completely covered by a shadow from trees or buildings. Meteorological events also affect this, such as heavy fog or rain. During the night, most of the light comes from the road infrastructure or the vehicle interior, which enables this sensor to be a good indicator of whether roads are well-illuminated or for identifying zones where ice is probable to be formed because of the lack of direct sunlight. Figure 18 shows a comparison between the illuminance values recorded from day and night scenarios.

The results show that, during the night, small variations occur with consistently low values. During the day, the opposite happens; the values vary between intermediate and high illuminance, causing it to be possible to clearly distinguish between the two scenarios.

It was also studied whether road hazards such as holes and humps could be detected by variations in the measures recorded by the inertial sensors of the smartphone. The setup involved placing the smartphone in a neutral position, with the screen turned in the direction of the driver. Figure 19 shows its orientation in the three-dimensional reference space.

In order to test this use case, a trial drive was performed mostly on a highway road where large speed humps were located at the beginning and at the end of the trip. Figure 20 shows the evolution of the acceleration of the smartphone in each axis during this trial.

With this setup, the *Y* axis corresponds to the vertical orientation, which is why it averages 1G (9.8 m/s2). With this orientation, the *X* corresponds to the horizontal orientation, increasing its values according to the sharpness of the turns performed, while the *Z* axis represents the depth, increasing its values with the actual acceleration of the vehicle.

While the acceleration values can be useful to report the status of the vehicle, or if the driver has adequate driving, they are not enough to indicate where street humps might exist. For this case, the gyroscope might be a good indicator, because the vehicle, and consequently the smartphone, rotates around the horizontal axis when passing over a hump. Figure 21 shows the evolution of smartphone rotation on each axis during this trial.

With the setup previously described, the values in the *Y* axis increase while the vehicle is turning, the values on the *Z* axis increase when one side (both left or right wheels) of the vehicle goes lower in altitude than the other side, and, finally, the values on the *X* axis increase when passing over a speed bump. Figure 22 shows only the rotation around this axis.

As expected, the module of the rotation measurements from the gyroscope presented bigger values around the beginning and the end of the trip, in accordance with the location of the big speed bumps. On the other hand, recordings of the pedometer evolution did not seem to have a correlation with any specific parameter on the road trip.

This approach can work with any smartphone because the *Expo-Sensors* process the raw measurements in a consistent way, with configurable sampling frequencies and using the same units. However, more research needs to be performed to understand if this is a good solution, as, in these tests, the smartphone was correctly positioned and standing throughout the whole journey might not be the case during normal use, where the phone might frequently change positions/orientations, causing the measured variations in each axis to be meaningless. A solution that covers this case could rely on the analysis of the absolute value/distance from the zero of the vector (x, y, and z). In this case, it is possible to work only with a single variable, causing it to be easy to identify variations but sacrificing the ability to distinguish between each direction. In the future, this feature for detecting risky road conditions could be enhanced or complemented with other state-of-the-art solutions based on similar sensors. For instance, the work of Gnap et al. The authors of [31] identify road locations with a higher risk of cargo damage, when the accelerometer from deployed inertial sensors reaches certain thresholds (0.5 g) for at least a minimum time interval, while Jagelčák et al. [32] use GNSS coordinates with acceleration data as an input in a statistical model in order to infer the turning radius and lateral acceleration forces.

Data redundancy is also useful if one wants to increase system availability (e.g., in case of failure when reading from one of the sensors) or improve the accuracy obtained in certain sensors’ measurements. Figure 23 shows a comparison between the same parameter (barometric pressure) between different sources, OBD-II and the smartphone, where it is clear that a higher data resolution is present in the smartphone data.

### 6.3. Communication and Data Layers Integration

Regarding the integration of the sensors with the other layers, the collected data are presented in a web interface to ease the interpretation by humans after being transmitted by the vehicles to a cloud platform. Figure 24 shows part of this interface. These data are sent via cellular networks to a cloud service that stores it in a database and provides an endpoint for historical data access. This endpoint is available on the PASMO API, in the *Vehicles* section (available at https://pasmo.es.av.it.pt/docs/, accessed on 1 February 2023; Example of query from 8th March 19:00 h–20:00 h (Aveiro to Oliveira de Azeméis): https://pasmo.es.av.it.pt/api/vehicles/69?initialDate=2022-03-08T19%3A00%3A00.000Z&finalDate=2022-03-08T20%3A00%3A00.000Z&limit=100, accessed on 1 February 2023).

## 7. Conclusions

The presented work proposes a solution to enrich the current RWMs and the C-ITS environment, taking advantage of the inherent mobility of each vehicle and their distribution through large geographical areas. The deployment of this solution is currently limited by the non-standardized nature of the content of CAN messages, as there is a need to configure each vehicle individually. This issue is somewhat mitigated by the usage of the OBD-II protocol, which standardizes the content of its messages, as well as with the usage of external sensors such as the ones available on smartphones. Another limitation is the fact that only light and medium duty vehicles (either conventional, electric, or autonomous) are required to possess an OBD-II connector. So, this solution cannot be deployed reliably in heavy vehicles as they are not required to have this interface, while some manufacturers optionally provide it and others have alternative diagnostic systems such as J1939, J1708, and J1587.

The extracted vehicular sensor data have various potential uses, for example, meteorological stats such as temperature and barometric pressure could be used to improve the current RWM based on stationary RWS by taking advantage of the inherent mobility of vehicles, enabling the identification of areas where ice is more probable to be formed due to low-temperature values, as well as inferring forest fire likelihood, as atmospheric pressure is proportional to ambient temperature and inversely proportional to humidity. It can also be used in extreme cases to detect forest fires, as barometric pressure suddenly drops in the vicinity of forest fires due to the pressure deficit created by oxygen consumption [33,34].

The analysis of the collected data could also be valuable to other drivers and connected cars, because it allows for the detection of anomalies in the traffic environment such as congestion, dangerous driving behavior, hydroplaning, predictive vehicle maintenance, or inadequate driving that causes high engine load values, directly influencing fuel consumption. By providing real-time notifications about important and dangerous events on the road, traffic safety and efficiency can be significantly improved. Furthermore, these data could also be used by authorities to identify zones more sensitive to traffic congestion, low visibility, accidents, and where drivers do not respect speed limits, indicating the need to reinforce surveillance mechanisms in these areas. Furthermore, statistical data on traffic flows are very valuable to marketing companies and can be used for selling points to consumers.

The mobile application described in the paper works as a proof of concept that takes advantage of the ETSI-defined services (e.g., CPS and VAS) in addition to the vehicular sensor data, which were disseminated using messages specifically created for the purpose. This provides an enhanced perception of the road environment, both for drivers and for VRU entities.

Since the architecture described is designed to be modular, extending this system to other data sources should be easy regardless of their radio access technology, as all that is needed is a handler to redirect the data to the MQTT broker, if it does not already support MQTT communication. Future iterations on this work include expanding the mobile application to use the smartphone’s camera, acquiring a CAN DBC for a specific vehicle in order to work with a ground truth set of parameters, performing more tests in various car models in specific and controlled conditions, and, finally, processing and analyzing the data, which can be fed into a machine learning model with the goal of profiling various geographical locations.

## Figures and Tables

**Figure 1 sensors-23-01724-f001:**
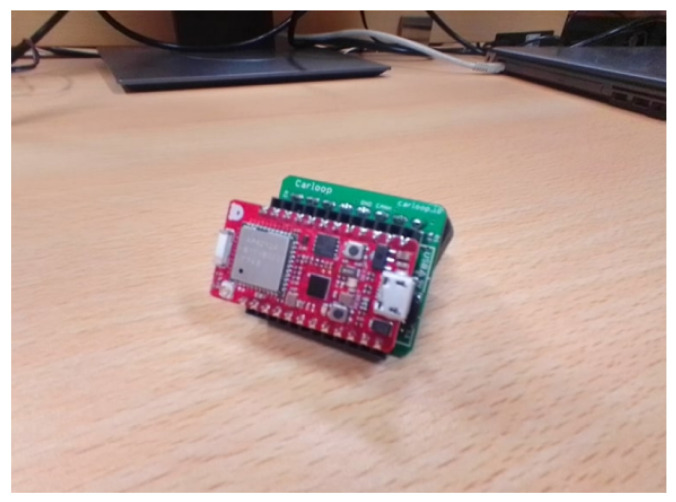
CAN transceiver and controller.

**Figure 2 sensors-23-01724-f002:**
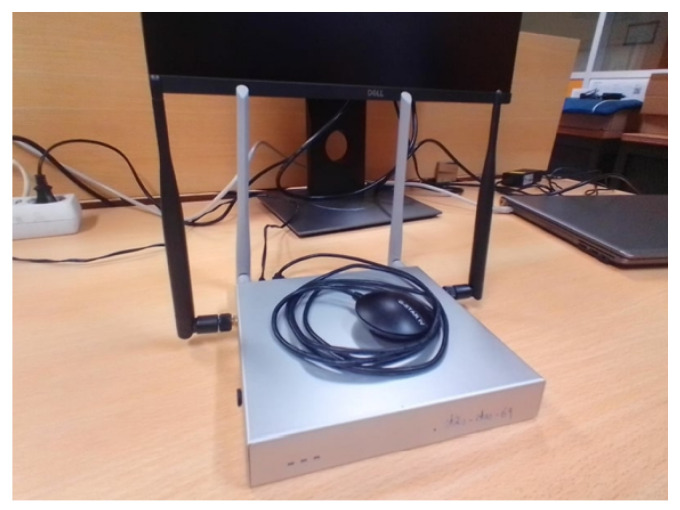
OBU with GPS receiver.

**Figure 3 sensors-23-01724-f003:**
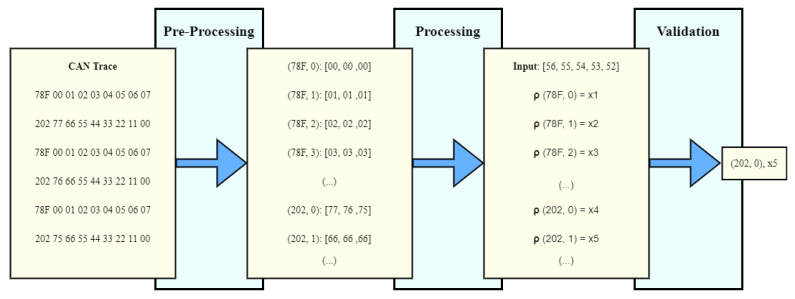
Representation of the proposed algorithm for CAN parameters’ identification.

**Figure 4 sensors-23-01724-f004:**
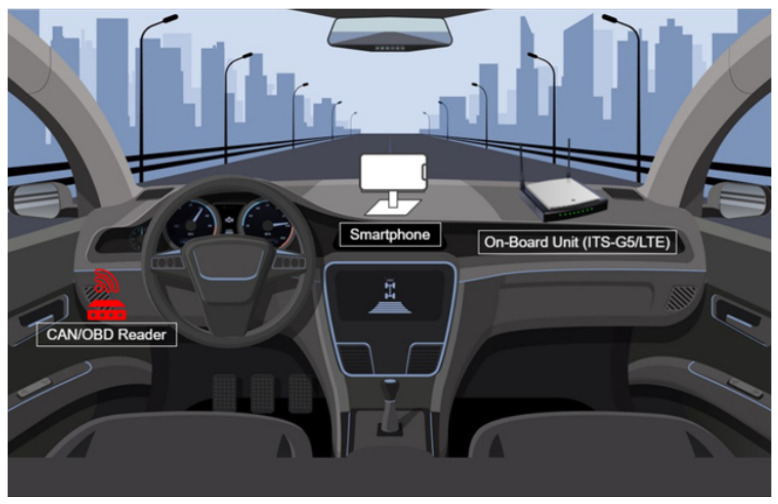
Representation of the devices installed in the vehicle.

**Figure 5 sensors-23-01724-f005:**
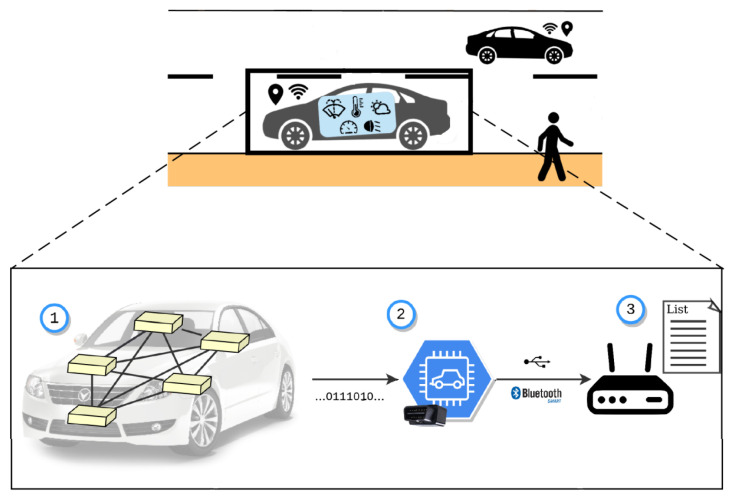
Representation of the vehicular data collection mechanism.

**Figure 6 sensors-23-01724-f006:**
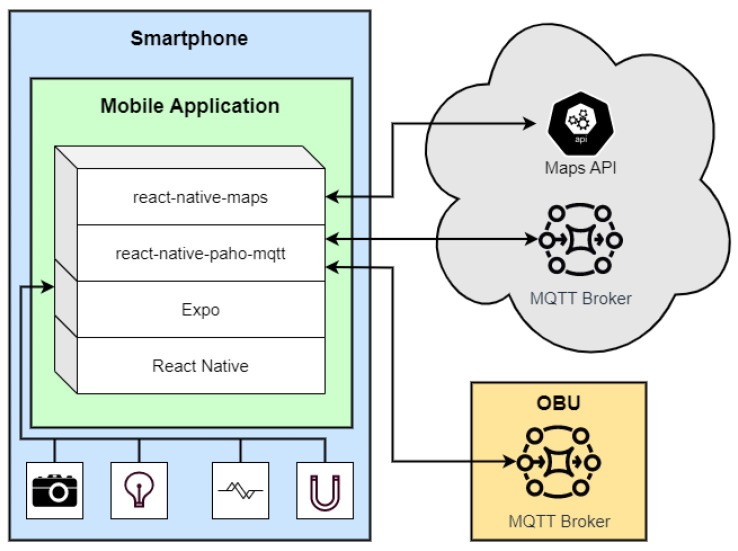
Diagram of the mobile application architecture.

**Figure 7 sensors-23-01724-f007:**
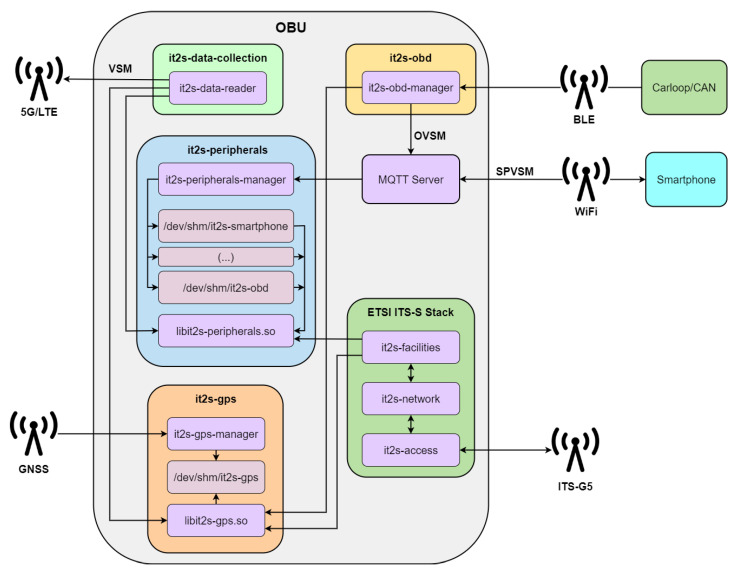
OBU’s internal architecture and interfaces.

**Figure 8 sensors-23-01724-f008:**
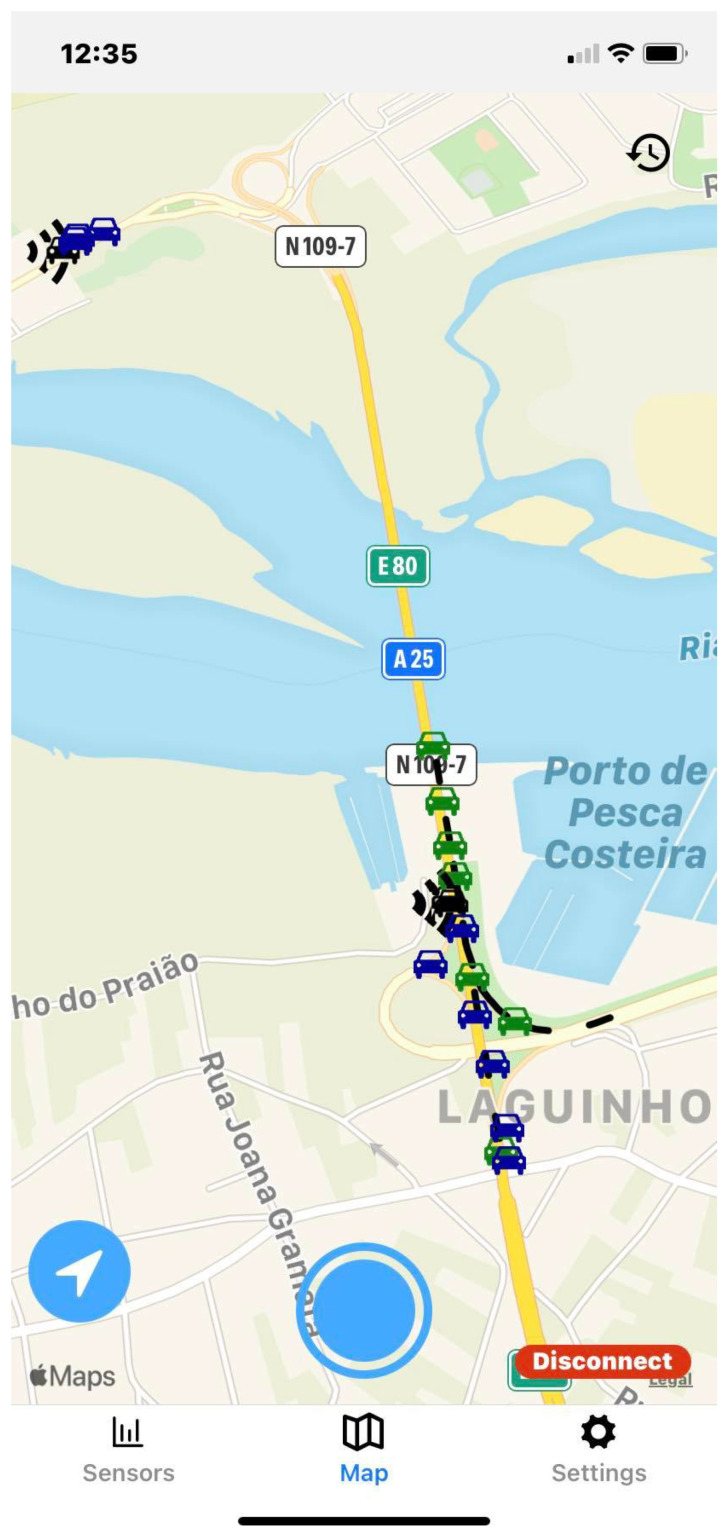
CAM and CPM message representation in the mobile app.

**Figure 9 sensors-23-01724-f009:**
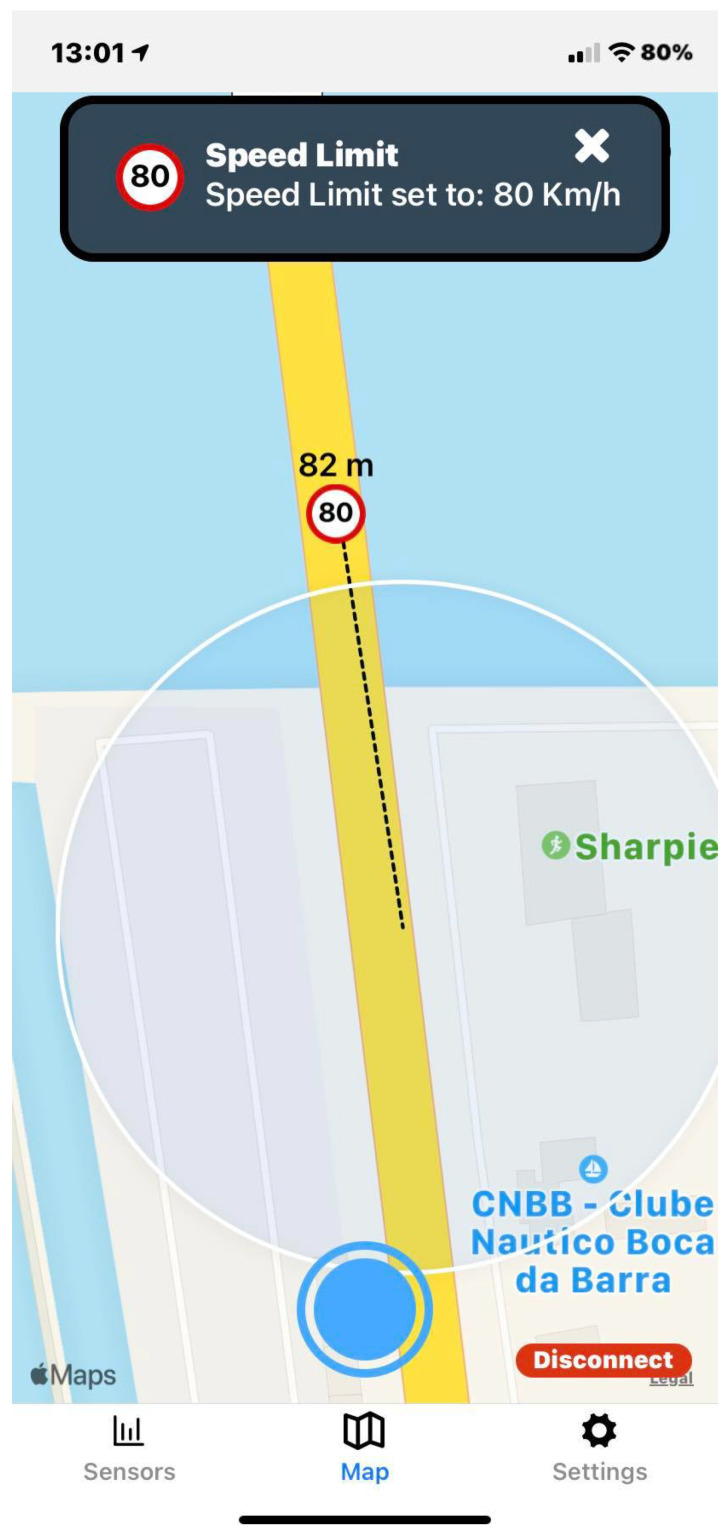
IVIM message representation in the mobile app.

**Figure 10 sensors-23-01724-f010:**
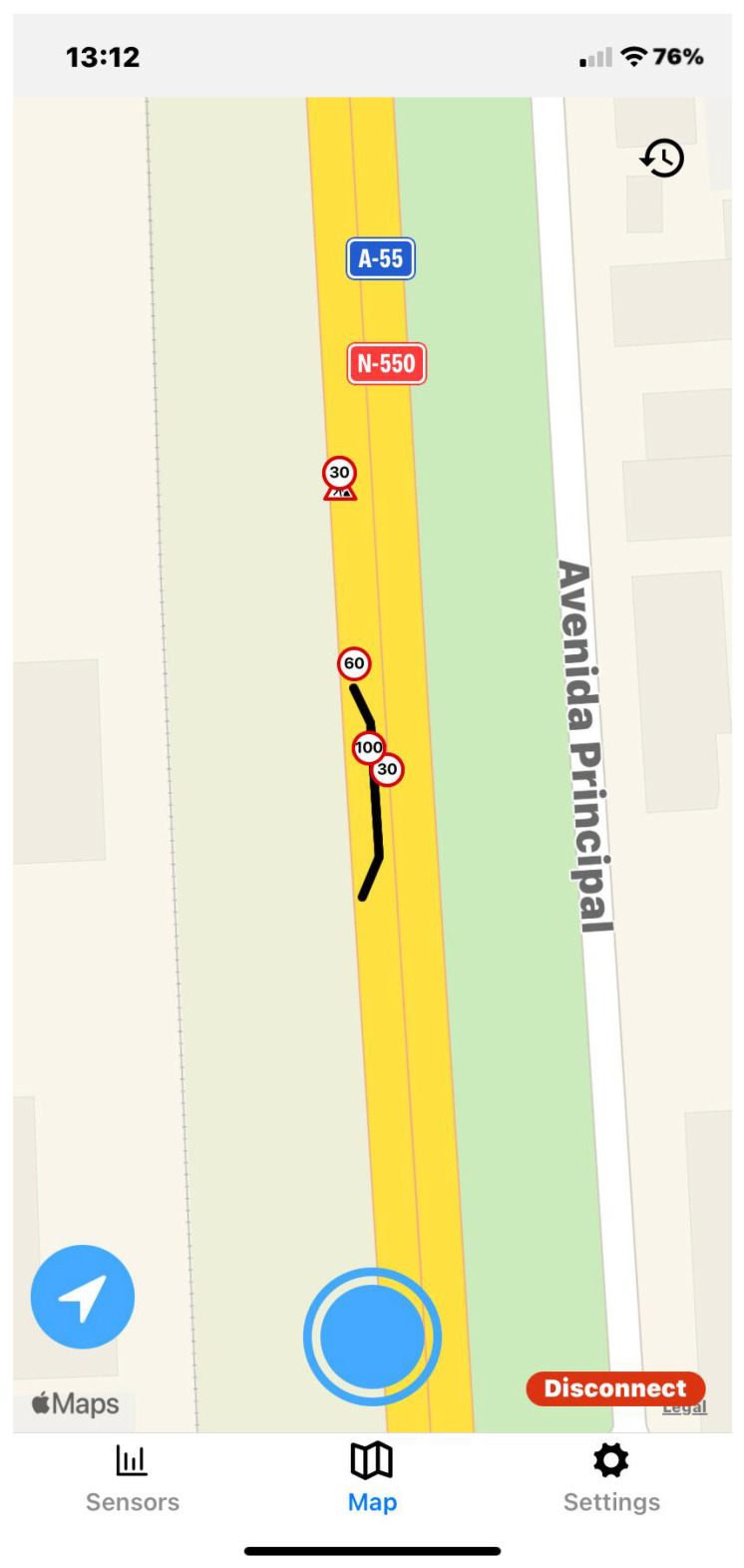
HD-Map message representation in the mobile app.

**Figure 11 sensors-23-01724-f011:**
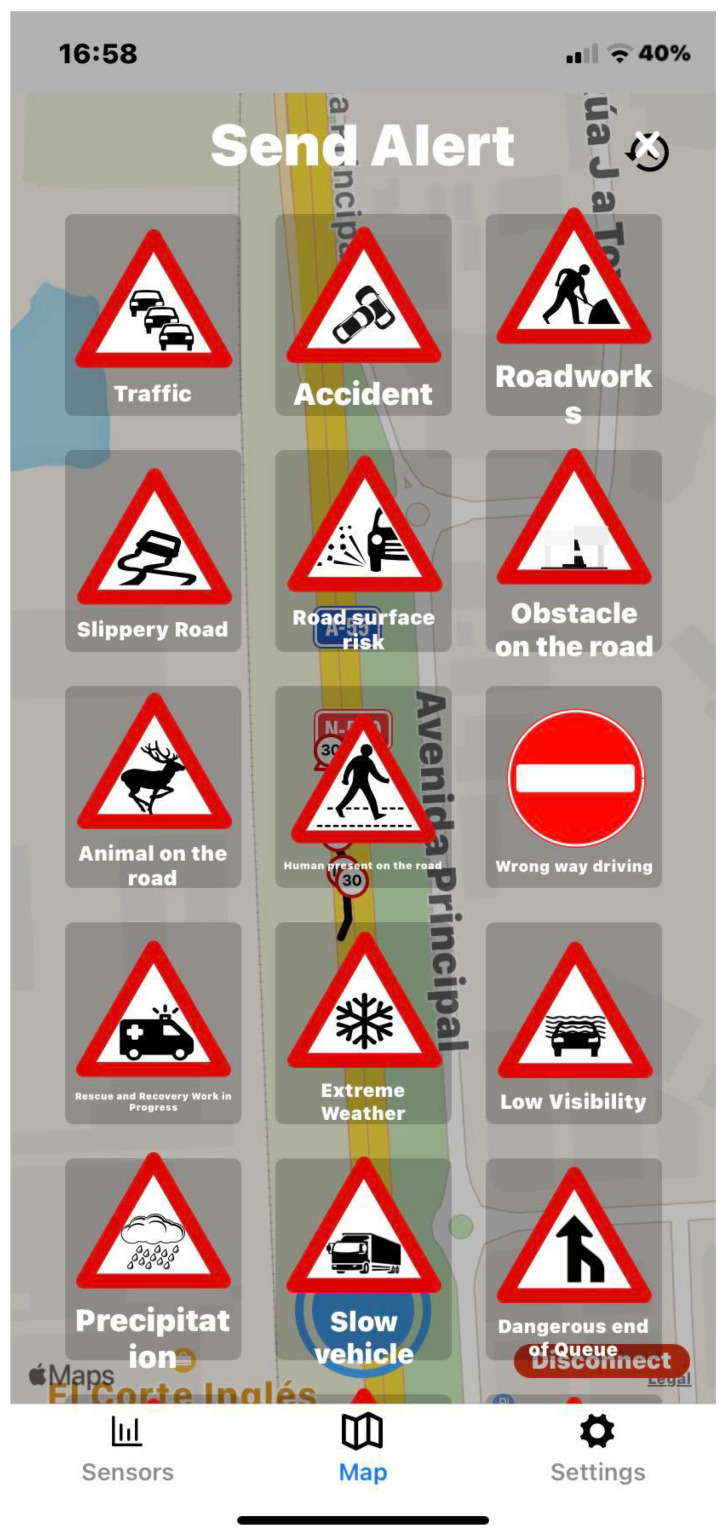
DENM message generation in the mobile app.

**Figure 12 sensors-23-01724-f012:**
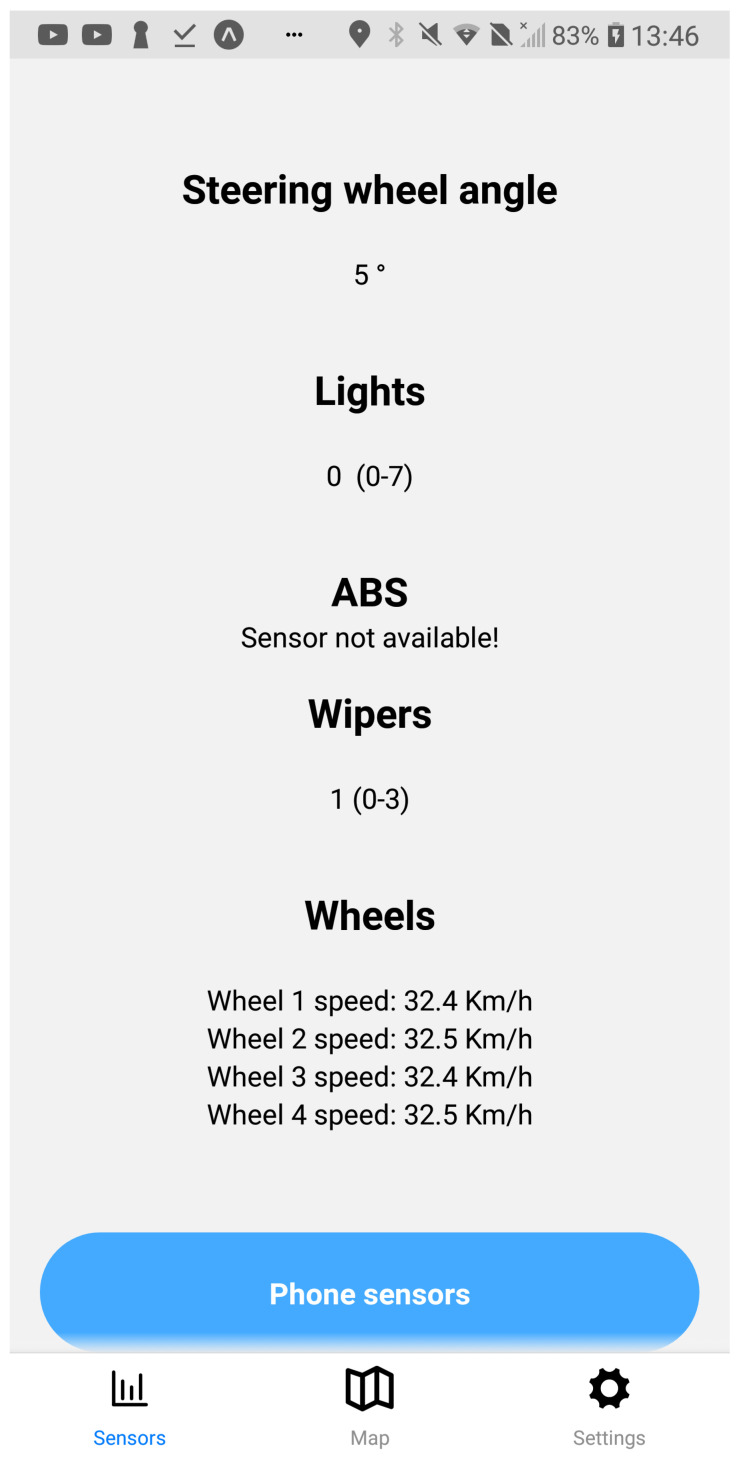
CAN/OBD-II’s sensor data representation in the mobile app.

**Figure 13 sensors-23-01724-f013:**
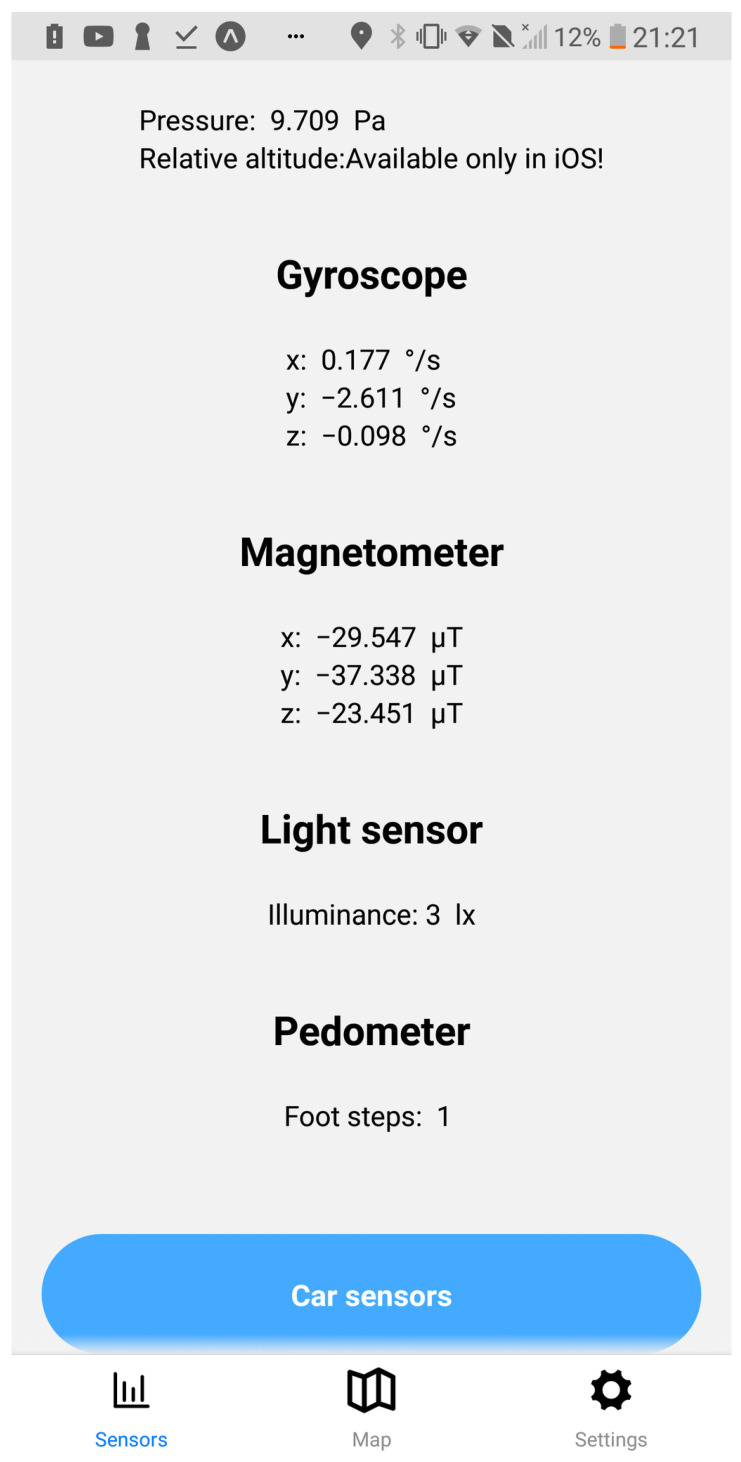
Smartphone’s sensor data representation in the mobile app.

**Figure 14 sensors-23-01724-f014:**
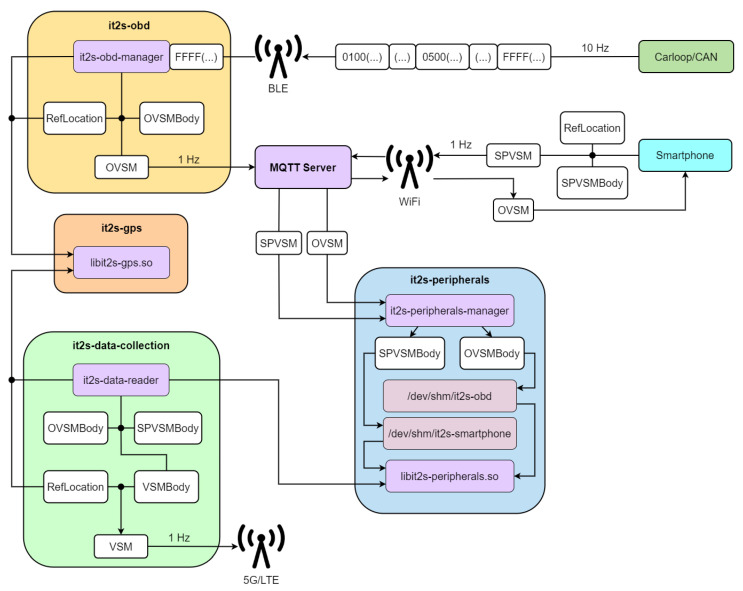
Dataflows between OBU’s services and peripheral devices (OBD-II reader and smartphone).

**Figure 15 sensors-23-01724-f015:**
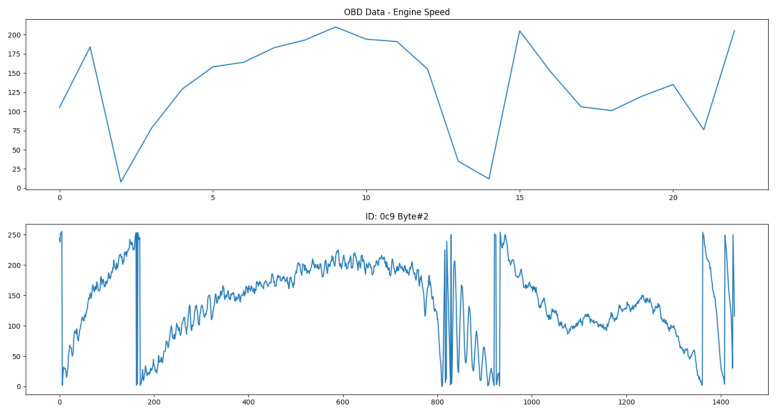
Second byte of OBD-II data for engine speed (**top**) and the best CAN candidate byte (**bottom**).

**Figure 16 sensors-23-01724-f016:**
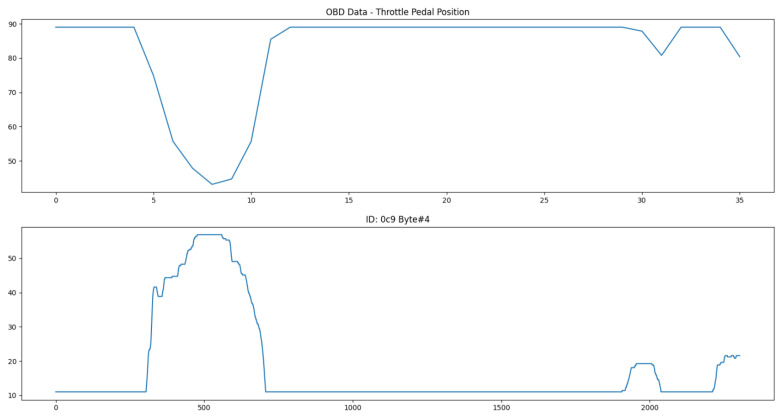
OBD-II data for throttle position (**top**) and the best CAN candidate byte (**bottom**).

**Figure 17 sensors-23-01724-f017:**
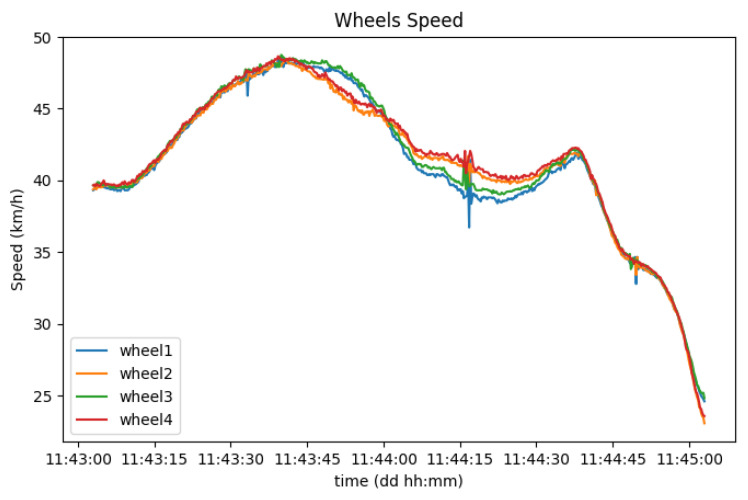
Variation of each wheel speed through a sharp turn.

**Figure 18 sensors-23-01724-f018:**
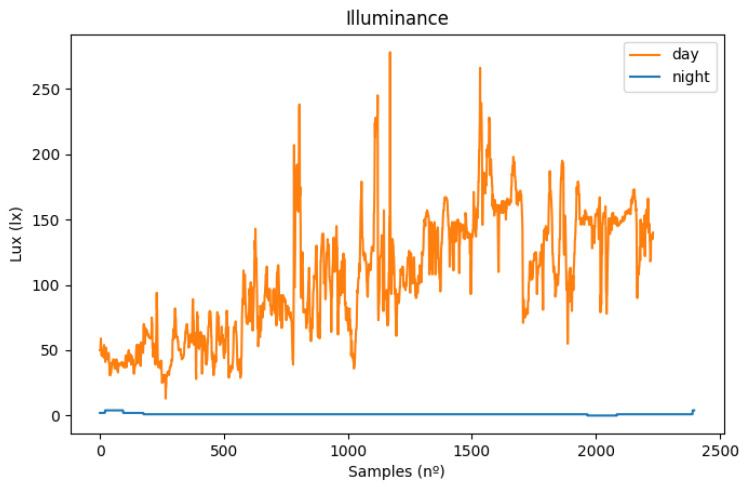
Illuminance values recorded during day hours (orange), and during the night (blue).

**Figure 19 sensors-23-01724-f019:**
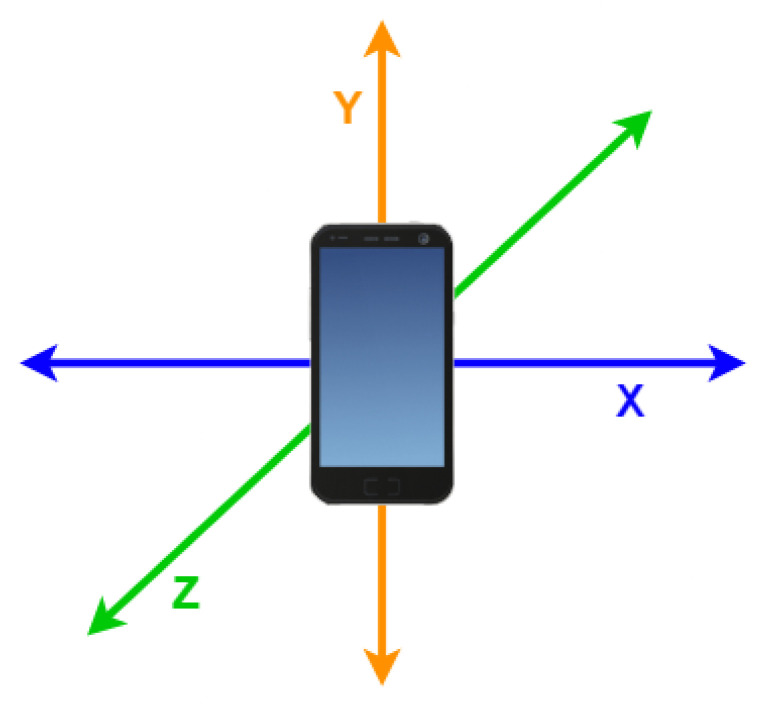
Orientation of the smartphone in the 3D reference space.

**Figure 20 sensors-23-01724-f020:**
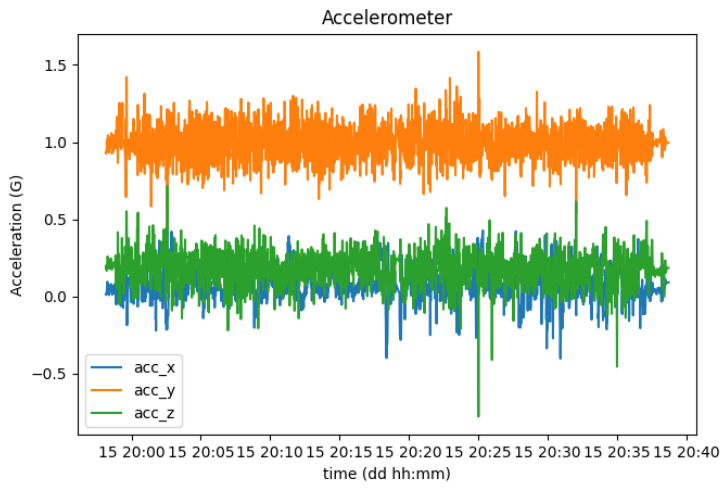
Acceleration of the smartphone in each axis (x, y, and z).

**Figure 21 sensors-23-01724-f021:**
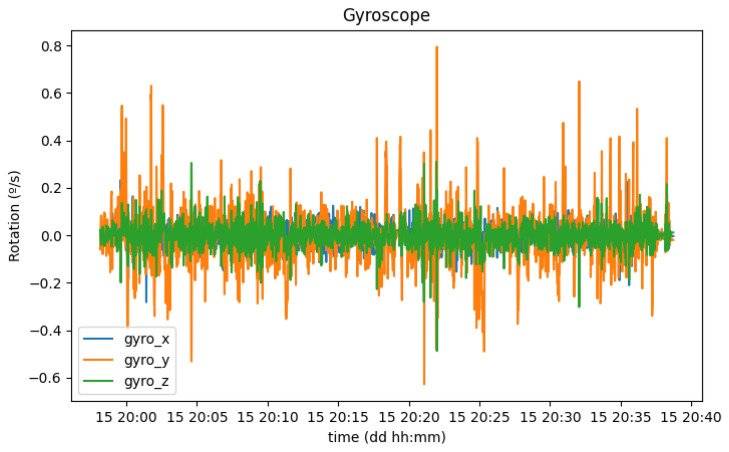
Rotation of the smartphone in each axis (x, y, and z).

**Figure 22 sensors-23-01724-f022:**
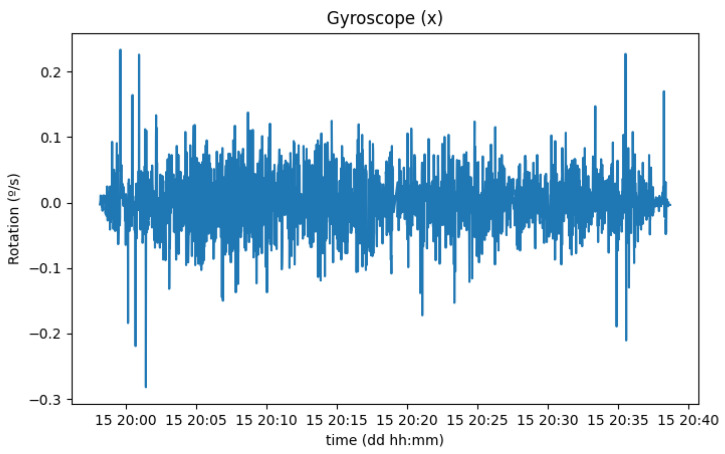
Rotation of the smartphone in the x-axis.

**Figure 23 sensors-23-01724-f023:**
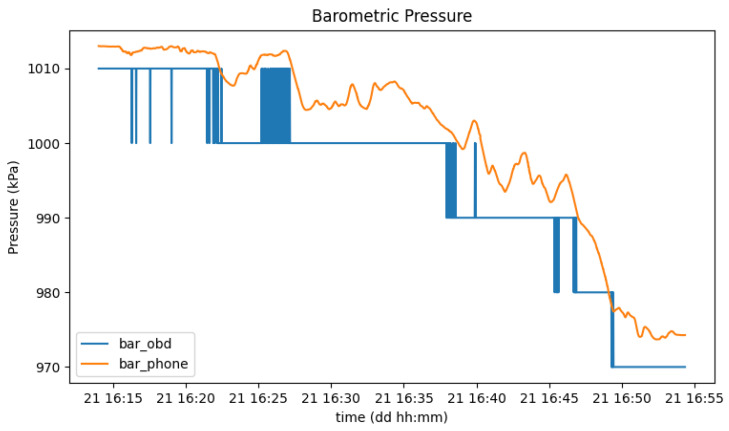
Barometric pressure measured from the OBD-II reader (blue) and the smartphone (orange).

**Figure 24 sensors-23-01724-f024:**
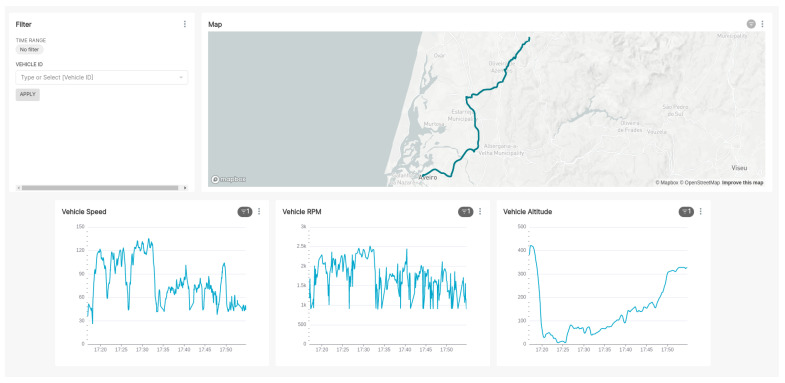
Web interface displaying the collected data.

**Table 1 sensors-23-01724-t001:** Overview of sensors available on test vehicles [20].

In Car	Sensor Box
GPS Module	Gyroscope
Camera	Accelerometer
Thermal Imaging Sensor	Temperature Sensor
	Humidity Sensor

**Table 2 sensors-23-01724-t002:** OBU platform specifications.

Motherboard Model	PC Engines apu3d4 = AMD GX-412TC CPU/4 GB
WiFi + Bluetooth Combo Adapter	VRK USB Wifi 5 and Bluetooth 5.0
Cellular Module	Huawei ME909s-120
ITS-G5 Module	Atheros Compex WLE200NX
GPS Receiver	G-Star IV

**Table 3 sensors-23-01724-t003:** Overview of sensors and parameters defined for data collection.

OBD-II	CAN	Smartphone
Vehicle Speed	Steering Wheel Angle	Gyroscope
Engine Speed	Wheels Speed	Accelerometer
Engine Load	Headlights Status	Magnetometer
Ambient Temperature	Wipers Status	Atmospheric Pressure
Throttle Pedal Position	Brake Pedal Position	Altitude
Atmospheric Pressure		Illuminance
		Pedometer

**Table 4 sensors-23-01724-t004:** Identified parameters on Opel Adam (2016) with its respective features.

Opel Adam (2016)	CAN ID (HEX)	Byte(s)	Unit	Offset	Scale
Engine Speed	0c9	1	rpm	0	1
Engine Load	3d1	1	%	−20	4/5
Vehicle Speed	3e9	0–1	km/h	0	1/64
Throttle Pedal	0c9	4	%	100	−1
Wheels Speed	34a	0–1	km/h	0	1/32
	34a	2–3	km/h	0	1/32
	348	0–1	km/h	0	1/32
	348	2–3	km/h	0	1/32

## Data Availability

The data presented in this study are available on request from the corresponding author. In any case, some of the data are already available using a public API (available at https://pasmo.es.av.it.pt/docs/, accessed on 1 February 2023).

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
