# Peer review of "A Modular In-Vehicle C-ITS Architecture for Sensor Data Collection, Vehicular Communications and Cloud Connectivity"

_sensors, 2023, doi:10.3390/s23031724_

Round 1

Reviewer 1 Report

Congratulations for the article written and for the field approached. Significant results and discussion are exposed however, some minor remark must be mentioned for improving the paper. 

Line 480-482: Is it possible to read outside temperature from OBD-II protocol or CAN messages?

Line 484: Seems that reference 24 could not be used to refer the influence of pressure on forest fires (liquid fire in indoor testing).

431-460,  How the raw data from MEMS accelerometer and MEMS gyroscope will be evaluated to detect bumps or other dangerous events (485-491) as for example in Application of MEMS Sensors for Evaluation of the Dynamics for Cargo Securing on Road Vehicles or Determination of Turning Radius and Lateral Acceleration of Vehicle by GNSS/INS Sensor?

Is it possible to use the proposed solution for heavy vehicles or vehicle combinations?

Reviewer 2 Report

The reviewer found this paper interesting and definitively the theme worthy of investigation. However, there are some aspects that require the authors’ attention, as follows:

1_ Since the issue is very popular, much more references could be added to strengthen your views. For example, there are no references cited in the first two paragraphs. Please reconsider. Contribution to road safety has been introduced by the concept of smart vehicle tires too (you may see and cite https://doi.org/10.3390/vehicles4030042).

2_ Lines 80-82: Please expand the impact of adverse weather conditions on the provided friction levels that needs to be communicated among individual vehicles to avoid collisions too.

3_ Improve the resolution and clarity of Figure 5.

4_ Lines 384-385: How is the developed approach validated in terms of training and accuracy? Please elaborate on the terms, traffic conditions, road geometry features, moving speeds, etc.

5_ Please add study limitations and future research directions.

6_ Provide a framework of how the developed sensors could be integrated within the current state of vehicle movements and mobility patterns. Are they to be integrated into the current conventional vehicles or their aim is only for the era of AVs? Please provide discussion points.

7_ There are too many figures in the text. You might exclude some of them (e.g., figure 2, 8, 9, 14, 15, 18).

Round 2

Reviewer 2 Report

No further comments.